

# On the importance of middle atmosphere observations on ionospheric dynamics using WACCM-X and SAMI3

Fabrizio Sassi[1,2], Angeline G. Burrell[1], Sarah E. McDonald[1], Jennifer Tate[3], and John P. McCormack[1,4]

[1]Naval Research Laboratory - Space Science Division, 4555 Overlook Ave SW, Washington DC 20375, USA
[2]Heliophysics Division, Goddard Space Flight Center, 8800 Greenbelt Rd., Greenbelt MD 20771, USA
[3]Computational Physics Inc., 8001 Braddock Rd, Suite 210, Springfield VA 22151, USA
[4]Heliophysics Division, Science Mission Directorate, 300 Hidden Figures Way SW, Washington DC 20546, USA

**Correspondence:** Angeline G. Burrell (angeline.burrell@nrl.navy.mil)

**Abstract.** Recent advances in atmospheric observations and modelling have enabled the investigation of thermosphere-ionosphere interactions as a whole atmosphere problem. This study examines how dynamical variability in the middle atmosphere (MA) affects day-to-day changes in the thermosphere and ionosphere. Specifically, this study investigates ionosphere-thermosphere interactions during different time periods of January 2013 using the Specified Dynamics Whole Atmosphere Community Cli-

mate Model, eXtended version (SD-WACCM-X) coupled to the Naval Research Laboratory (NRL) ionosphere of the Sami3 is Another Model of the Ionosphere (SAMI3) model. To represent the weather of the day, the coupled thermosphere-ionosphere system is nudged below 90 km toward the atmospheric specifications provided by the Navy Global Environmental Model for High-Altitude (NAVGEM-HA). Hindcast simulations during January 2013 are carried out with the full data set of observations normally assimilated by NAVGEM-HA, and with a degraded dataset where observations above 40 km are not assimilated.

Ionospheric regions with statistically significant changes are identified using key ionospheric properties, including the electron density, peak electron density, and height of the peak electron density. Ionospheric changes show a spatial structure that illustrates the impact of two different types of coupling between the thermosphere and the ionosphere: wind-dynamo coupling through electric conductivity and ion-neutral interactions in the upper thermosphere. The two simulations presented in this study show that changing the state of the MA affects ionosphere-thermosphere coupling through changes in the behavior and

amplitude of non-migrating tides, resulting in improved key ionospheric specifications.

## 1  Introduction

The development of whole atmosphere models (ground to exobase) in the last decade along with the availability of high-altitude atmospheric data assimilation has brought the geospace community closer to important understandings of the prediction of short-term variability ($\leq 10$ days) in the lower E-region. When the Sun is quiescent, a large fraction of this short-term variability

is driven by lower atmosphere weather from below. Because of wind-dynamo coupling in this region, neutral variability affects the state of the ionosphere both locally and at higher altitudes.

The impact of atmospheric weather on the behavior of the coupled thermosphere-ionosphere system is a topic that has been actively investigated for nearly a century. From small-scale gravity waves (Hines, 1960) to planetary-scale waves (Forbes and



Zhang, 1997), meteorological weather impacts the ionosphere either through electrodynamics (e.g., $\mathbf{E} \times \mathbf{B}$ drifts) or collisional
interactions whose impacts greatly affect ion transport along field lines. Forbes et al. (2000) show that the fraction of iono-
spheric variability (obtained from the critical frequency at the F-peak, *foF2*) due to meteorological conditions depends on the
period of the meteorological forcing itself, and varies between 20% and 35% during solar minimum conditions; modeling
studies tend to estimate a higher fraction for the meteorological contribution, approaching 50% (e.g. Liu et al., 2013). The role
of migrating and non-migrating tides in determining the variability and structure of the ionosphere on many time scales, from
days to seasons, is prominent and well established (Hagan et al., 2007; Liu, 2016; Sassi et al., 2019, and references therein).
Migrating solar tides are associated with a modulation of the daily variability of vertical ion drifts at the geomagnetic equator
(Millward et al., 2001; Fang et al., 2013). In particular, the migrating semidiurnal solar tide has been associated with a shift of
meridional ion drifts (which are directed vertically at the geomagnetic equator), and consequently of the peak electron density,
to earlier local times during days immediately following a Sudden Stratospheric Warming (SSW) (e.g., Goncharenko et al.,
2010; McDonald et al., 2015; Fuller-Rowell et al., 2017). Non-migrating tides have instead been associated with the zonal
structure of the ionosphere (Forbes et al., 2008; Immel et al., 2006) and have been found to be very sensitive to the background
meteorological conditions of the lower atmosphere (McDonald et al., 2018; Sassi et al., 2020). In addition to solar tides, lunar
tides are known to be impactful on the field-aligned neutral winds that help shape the structure of the F-peak ionosphere (Pe-
datella and Maute, 2015). Furthermore, a statistical analysis that aggregates the response of the Equatorial Ionization Anomaly
(EIA) over many SSW (Wu et al., 2021) shows that the timing of the modulation of the EIA following an SSW is affected by
the lunar phase.

Such theoretical progress in understanding the coupling between the thermosphere and the ionosphere has, in part, been
possible because of momentous advances in the development of whole atmosphere numerical models during the last decades
that also include electrodynamic interactions (e.g., Liu et al., 2010, 2018). At the same time, the development of these modeling
capabilities has benefited from the availability and utilization of observations well into the upper atmosphere and the extension
of forecast/assimilation systems that provide specifications in the upper atmosphere (Eckermann et al., 2009; Wang et al.,
2011; McCormack et al., 2017; Eckermann et al., 2018). Thus, investigations of the Coupled Ionosphere Thermosphere System
(CITS) have been guided and validated by observations (Jin et al., 2012), both through nudging techniques (e.g., Sassi et al.,
2013) and full data assimilation (Wang et al., 2011; Pedatella et al., 2018). In most cases, the need for upper atmospheric
observations is not limited to merely validating forecast and hindcast simulations, but rather extends to fundamental advances
in theoretical understanding of whole atmosphere interactions (Liu, 2016; Sassi et al., 2019, for reviews). Observations are
critical to evaluate and improve forecast skill in the whole atmosphere: Liu et al. (2009) demonstrated that the troposphere
plays a critical role in controlling error growth at higher altitudes. Pedatella et al. (2013) used a data assimilation system to
reduce the error growth up to 40% in the Upper Mesosphere and Lower Thermosphere (UMLT) and in a later study, Pedatella
et al. (2019) used a whole atmosphere model with a data assimilation system to demonstrate that the error growth in the lower
thermosphere saturates within 5 days.

Observations of the middle atmosphere (MA), which consists of both the stratosphere and mesosphere, are primarily ob-
tained from instruments carried onboard space-borne platforms. Unfortunately, many research platforms have surpassed their





expected lifetime, such as the Sounding of the Atmosphere using Broadband Emission Radiometry (SABER) instrument on-
board the NASA Thermosphere Ionosphere Mesosphere Energetic Dynamics (TIMED) satellite, while critical operational
platforms are not being replaced (e.g., the Defense Meteorological Satellite Program (DMSP)) (e.g., Erwin and Berger, 2021).
As noted by Sassi et al. (2020), the potential lack of observations in the MA has significant implications for the day-to-day
variability of neutral winds in the lower thermosphere. In particular, the amplitude of some non-migrating solar tides and of
traveling planetary scale waves can change by a factor of two. It stands to reason that such prominent changes of the thermo-
spheric weather will have consequences for the structure and variability of the ionosphere. To the best of our knowledge the
fundamental question of how numerical simulations of physical properties of the CITS are impacted by data loss in the middle
atmosphere remains elusive and poorly described.

This study seeks to answer the fundamental question of whether the loss of MA observations is potentially consequential for
predictions of thermosphere-ionosphere coupling. In Section 2, the different models used in this study are presented, and details
about the run configurations are provided. Section 3 presents the results of the model runs, Section 4 presents a discussion of
the results and Section 5 the conclusions.

## 2   Models

Three models are used in this study: the Navy Global Environmental Model (NAVGEM-HA), the Whole Atmosphere Com-
munity Climate Model, eXtended version (WACCM-X), and the Sami3 is Another Model of the Ionosphere (SAMI3). Details
about these models and how they have been integrated to effectively model the CITS are presented below.

### 2.1   NAVGEM-HA

The Navy Global Environmental Model (NAVGEM) is the Navy's operational global numerical weather prediction (NWP)
system. NAVGEM couples a semi-implicit semi-Lagrangian atmospheric model with a hybrid four-dimensional variational
(4DVAR) data assimilation system that is based on the NRL Atmospheric Variational Data Assimilation System–Accelerated
Representer (NAVDAS-AR) (Kuhl et al., 2013). This study uses a version of NAVGEM developed especially for High Alti-
tude research (NAVGEM-HA). The standard version of NAVGEM only covers 0-80 km, but NAVGEM-HA has an extended
altitude range that reaches 116 km and includes additional physical parameterizations for high altitude processes such as
ozone and water vapor photochemistry, as well as sub-grid scale gravity wave drag. For a detailed description, see McCor-
mack et al. (2017) and Eckermann et al. (2018). In addition to the full suite of operational observations and space-based
sensor data, NAVGEM-HA assimilates temperature profiles between 30 and 100 km altitude (provided by the TIMED SABER
instrument), temperature, ozone, and water vapor profiles (provided by the Aura Microwave Limb Sounder), and Special Sen-
sor Microwave Imager/Sounder (SSMIS) Upper Atmospheric Sounding (UAS) channel radiances (Hoppel et al., 2013). The
standard NAVGEM-HA configuration has T119 triangular spectral resolution (e.g., Laprise, 1992) (1° latitude/longitude grid
spacing), with 74 vertical levels (L74) having an effective spacing of ∼2 km in the stratosphere, ∼3 km in the mesosphere
and ∼4 km in the lower thermosphere. The present study uses the NAVGEM-HA global synoptic analyses of key atmospheric



state variables (including temperature, winds, geopotential height, ozone, water vapor, vorticity, and divergence) every 6 hours. These atmospheric state variables have been augmented with 3-hour forecasts to provide 3-hourly output. This greater output frequency is needed to resolve sub-diurnal variability that is crucially important to represent the day-to-day variability in the upper atmosphere and ionosphere.

Two sets of NAVGEM-HA atmospheric specifications have been generated for January-February 2013. The first is a reference set with all observations from the ground to 100 km; this hybrid data assimilation with MA observations is referred to as *hybma* (*hyb*rid *MA*). A second set of NAVGEM-HA simulations have been generated excluding all observations above 40 km; this theory-based set is referred to as *noobs* (*no* MA *obs*ervations). Temperature, winds and surface pressure generated by *hybma* and *noobs* NAVGEM-HA are used to nudge the atmosphere of SD-WACCM-X (see Sect. 2.2). This is exactly the same

dataset of atmospheric specifications that has been used in the Sassi et al. (2020) study.

## 2.2    SD-WACCM-X

To determine the neutral atmospheric response to the weather of the day from the lower atmosphere throughout the thermosphere, we use the Whole Atmosphere Community Climate Model, eXtended version (WACCM-X; Liu et al. (2010)). WACCM-X horizontal resolution is $\sim 2°$ in latitude and longitude. The vertical resolution, while variable in the lower atmo-

sphere, is set to be a quarter of the local pressure scale height in the thermosphere, with 108 vertical levels from the ground to about 500 km. To aid in the investigation of whole atmosphere connections, WACCM-X can be configured to use atmospheric specifications to constrain its meteorology (winds and temperature) from the ground to any altitude; this model configuration is referred to as Specified Dynamics (SD). The reader is referred to Sassi et al. (2020) for an exhaustive discussion of the day-to-day variability of these simulations.

As in Sassi et al. (2020), the *hybma* and *noobs* atmospheric specifications generated by NAVGEM-HA are used to nudge the SD-WACCM-X meteorology. In the remainder of this article, except when stated otherwise, *hybma* and *noobs* are used as abbreviations to refer to the SD-WACCM-X simulations with the corresponding NAVGEM-HA nudging meteorology.

## 2.3    SAMI3

SAMI3 is a longitudinal extension of SAMI2 (Huba et al., 2000), a two-dimensional model of the ionosphere. SAMI3 models

the plasma and chemical evolution of seven ion species ($H^+$, $He^+$, $N^+$, $O^+$, $N_2^+$, $NO^+$ and $O_2^+$) in the altitude range extending from 70 km to $\sim 8$ Earth radii and magnetic latitudes up to $\pm 88°$. Unlike other ionospheric models, SAMI3 solves the ion continuity and momentum equations for all seven of the ion species, and solves the temperature equations for the electrons and three ion species ($H^+$, $He^+$ and $O^+$). Ion inertia is included in the ion momentum equation for motion along the geomagnetic field. SAMI3 is the only global ionosphere code that models full plasma transport for all of these ion species, including

the molecular ions. Another unique feature of SAMI3 is its non-uniform, nonorthogonal fixed grid. The original version of SAMI3 used a tilted dipole model of the geomagnetic field. However, SAMI3 has been upgraded to allow use of Magnetic Apex coordinates (Richmond, 1995), so that an accurate representation of the geomagnetic field provided by the International Geomagnetic Reference Field (IGRF-13) is considered. SAMI3 includes a potential solver to self-consistently solve for the





electric fields. The potential equation is derived from current conservation in magnetic coordinates, using Ohm's law and including gravity-driven currents. The perpendicular electric field is used self consistently in SAMI3 to calculate the $\mathbf{E}\times\mathbf{B}$ drifts driven by the neutral wind in the low- to mid-latitude ionosphere (Huba et al., 2010). At high latitudes SAMI3 uses an empirical model (e.g., Weimer, 1995, 2005), and the solar extreme ultraviolet (EUV) irradiance is determined from the EUVAC model, as described in (Solomon and Qian, 2005). The software infrastructure that extends SAMI3 capability to be fully interactive with the atmosphere consists of interpolation software of the Earth System Modeling Framework (ESMF, https://earthsystemmodeling.org/), and an extension of neutral field properties to the plasmasphere (McDonald et al., 2018).

In this study we use the SD-WACCM-X winds along with the NRLMSIS 2.0 (Emmert and Co-authors, 2021) temperature and composition to define SAMI3 neutral properties. We chose this set up (similar to the one used in McDonald et al. (2015)) because we want to isolate as much as possible the effects of neutral winds from other factors; a more realistic and nonlinear system that includes the influences of meteorological winds, temperature and composition is bound to be far more complex and challenging to isolate the effects of neutral winds. The output cadence of the SAMI3 fields used for the analysis described in this study is 15 minutes.

## 3 Results

This section presents three local time (LT) cases: two cases during a dynamically active week (1–10 January), one in the morning (10:00 LT) and one in the afternoon (16:00 LT) and an afternoon (16:00 LT) case during a dynamically quiet time (21–30 January). Together, they illustrate different paths by which the neutral atmosphere may affect the ionosphere. The different paths will be examined for both *hybma* and *noobs* simulations (see Section 2.2). The first case (Section 3.1) is a morning scenario showing a longitudinally complex behavior with changes of both the lower thermospheric neutral wind and the F-peak winds that prominently affect the ionosphere through wind-dynamo coupling in the E-region causing electrodynamic interactions in the F-region (Heelis, 2004) and modulation of the EIA from inter-hemispheric ion transport at the F-peak, respectively. The second case (Section 3.2) demonstrates that the different specification of the MA state also leads to changes in the thermospheric winds in the F-region, which interact collisionally with the ions and alter the ion transport along the magnetic field lines (e.g., Rishbeth, 1972; Burrell and Heelis, 2012). The third case shows that the MA state affects the ionosphere and thermosphere during dynamically quiet times as well as dynamically active times.

### 3.1 Case 1: 10:00 LT - dynamically active

Figure 1 illustrates the $N_mF_2$ at 10:00 LT obtained from averaging the 15-minute output for ten days between January 1 and January 10, 2013, i.e., covering the time period immediately prior and following the SSW of January 6, 2013. In each panel, the location of the magnetic equator is marked by a solid black line and the longitude sector of interest is marked by a dashed black line. Both model runs show that the structure of the $N_mF_2$ is geomagnetically oriented, as the peak values at each longitude lie just north of the magnetic equator. In *hybma* (Figure 1a) the $N_mF_2$ is broadly uniform north of the magnetic equator at about $175\times10^4 - 200\times10^4 cm^{-3}$, with a relative maximum near 80°E just exceeding $200\times10^4 cm^{-3}$. In contrast, the $N_mF_2$





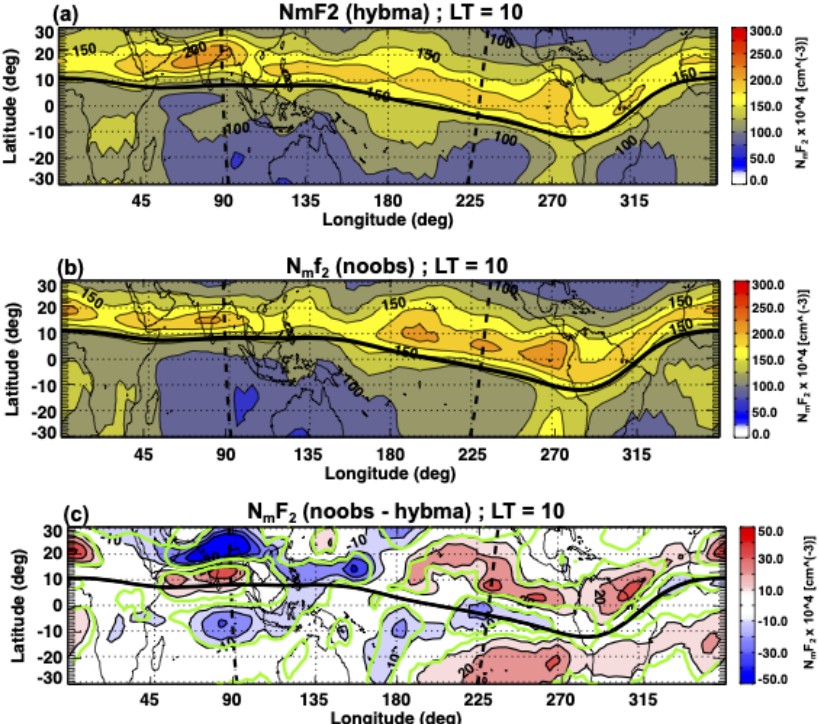

**Figure 1.** $N_mF_2$ in units of $10^4 \text{cm}^{-3}$ for all locations at 10:00 LT averaged for 1-10 January 2013: *(a) hybma*; *(b) noobs*; *(c)* difference *noobs* minus *hybma*. The oblique dash black lines highlight the position of SAMI3 slices whose longitude at the geographic equator are $\sim230°$ and $\sim90°$ and are discussed in text. The magnetic equator is illustrated by a solid black line across longitude. Green contours in *(c)* identify the locations where the anomalies are statistically significant at least at 95% confidence level using a t-test.

in *noobs* (Figure 1b) shows a more longitudinal structure north of the magnetic equator. The $N_mF_2$ displays several peaks in longitude, all of about the same magnitude and just exceeding $200 \times 10^4 \text{cm}^{-3}$. The longitudinally dependent changes are better illustrated by examining the difference between the model results. Figure 1c shows *noobs* minus *hybma* $N_mF_2$. The difference in peak density clearly shows that the largest effects introduced by removing MA observations through the *noobs* run at this

LT are over the Pacific Ocean and over the Indian Ocean. At the magnetic equator the differences in the two longitudinal regions have opposite signs: a negative anomaly flanked by positive differences in the Pacific Ocean, contrariwise over the Indian Ocean. Overall, the magnitude of the differences are up to $\pm50 \times 10^4 \text{cm}^{-3}$. Such longitudinal structure at a fixed LT introduced by changes to the lower atmospheric forcing can only be explained by differences in amplitude of non-migrating solar tides (Forbes et al., 2006).

Focusing on the region over the Pacific Ocean near 225°E, this region shows statistically significant differences between the *noobs* and *hybma* runs: a decrease of $N_mF_2$ at the magnetic equator is flanked by increases in the $N_mF_2$ at latitudes corresponding to the peak of the EIA. The symmetry of this structure about the magnetic equator implies a change to the





'fountain effect' (e.g., Hanson and Moffett, 1966): the EIA is maintained through $\mathbf{E} \times \mathbf{B}$ drifts that lift to higher altitudes ions created at or near the magnetic equator at lower altitudes. Opposed by gravity, the $\mathbf{E} \times \mathbf{B}$ vertical lift runs out near the

F-peak where the ions then diffuse downwards along magnetic field lines to lower altitudes and higher latitudes. To verify that a stronger fountain effect is taking place at this location, Figure 2 shows the electron density on nearby SAMI3 slice (longitude at the geographic equator $\sim 230°$) that cuts through the region of the largest differences (see oblique dash line in Figure 1). The electron density in *hybma* (Figure 2a) shows the typical arced behavior, with isopleths elevated near the magnetic equator with respect to the adjacent latitudes. The largest electron density is found just below 300 km northward of

the geographic equator, with a peak value in excess of $180 \times 10^4 cm^{-3}$. This is consistent with the presence of both an active fountain effect and a northward field-aligned neutral wind. The difference plot (Figure 2b) shows reduced electron density by about $15\text{-}30 \times 10^4 cm^{-3}$ at the magnetic equator, and increased values in excess of $60 \times 10^4 cm^{-3}$ around 5–10°N. This structure is explained by an increased uplift of ions at the magnetic equator followed by transport along geomagnetic field lines due to the effects of neutral winds and gravity. In both simulations, $\mathbf{E} \times \mathbf{B}$ meridional drifts at the magnetic equator lift ions into the

upper F-region; in the case of the *noobs* simulation, a stronger uplift of ions followed by gravity and neutral wind transport results in a removal of charges from the magnetic equator and an accumulation at higher latitudes. Note that the accumulation of charges is greater in the northern hemisphere, which (at this local time) is likely solely the result of forcing by the neutral wind at the $h_mF_2$ along these magnetic field lines (Burrell et al., 2012).

If our interpretation of Figure 2 is correct, the *noobs* simulation ought to be showing a more prominent upward drift in the

morning hours when compared to *hybma*. Figure 3 shows the 10-day average (*(a,b)* 1–10 January 2013) of the SAMI3 meridional $\mathbf{E} \times \mathbf{B}$ drift at the magnetic equator as a function of longitude and LT. At longitudes around 230°E (which corresponds to the location of SAMI3 slice in Figure 2 ) and at 10:00 LT, Figure 3 shows larger $\mathbf{E} \times \mathbf{B}$ drifts in *noobs* than in *hybma*. Since the resulting meridional $\mathbf{E} \times \mathbf{B}$ drift is upward during the day, the electric field must be directed toward the East, resulting in an accumulation of positive (negative) charges at dusk (dawn). Thus, our model results illustrate that the simulation without

MA observations (*noobs*) produces a stronger eastward electric field compared to the simulation with full MA observations, accumulating more charges at the dusk/dawn sectors. The stronger upward $\mathbf{E} \times \mathbf{B}$ drifts lift the F-region plasma to higher altitudes where they then diffuse downwards and polewards along the magnetic field lines. Figure 3 results then confirm that the larger EIA seen in Figure 1c is due to a stronger fountain effect, which was created by unrealistic influences from the MA in the *noobs* experiment.

We now turn to the large differences over the Indian Ocean in Figure 1. The resulting difference field at this longitude (Figure 1c) shows an increase of $N_mF_2$ just north of the magnetic equator flanked by negative anomalies in both hemispheres. Overall, there is a hemispheric asymmetry, with the largest differences clearly in the Northern Hemisphere. This difference pattern is the opposite of the one shown at the longitude over the Pacific Ocean. There are two mechanisms that may explain these differences: a difference of the meridional $\mathbf{E} \times \mathbf{B}$ drifts resulting in a decrease of the drifts in *noobs*, and a change to

the neutral wind driven ion transport at the F-peak. Referring to Figure 3(a,b) we notice that at this longitude ($\sim 90°$) the meridional $\mathbf{E} \times \mathbf{B}$ in the *noobs* simulation is smaller than in *hybma*. Figure 4 shows the electron density on the SAMI3 slice over the Indian Ocean in *hybma* (Figure 4a) and the differences *noobs* minus *hybma* (Figure 4b). The electron density in the



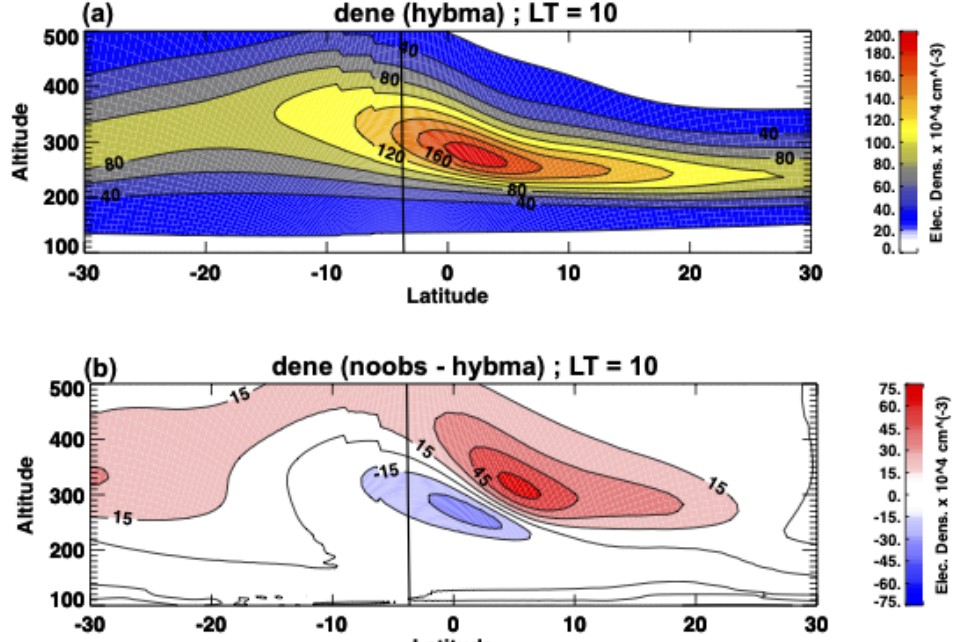

**Figure 2.** Electron density (dene) at 10:00 LT in units of $10^4 \text{cm}^{-3}$ along SAMI3 slice whose longitude at the equator is $\sim$230° averaged for the same period as in Figure 1. The vertical black line identifies the magnetic equator. *(a)* electron density in *hybma*; *(b)* difference *noobs* minus *hybma*.

reference simulation (*hybma*) shows the typical arced structure with elevated isopleth of density at the magnetic equator and lowered isopleths in the subtropics: they result from the competing uplift of ions by the meridional $\mathbf{E}\times\mathbf{B}$ drifts at the magnetic

equator and gravity. The difference of electron density between *noobs* and *hybma* just north of the magnetic equator agrees with the $N_m F_2$ differences in Figure 1. In this case, a weakening of the fountain effect via $\mathbf{E}\times\mathbf{B}$ drifts at this longitude (Figure 3a,b) creates the pattern in Figure 1c and Figure 4: both the positive and negative anomalies are necessary to explain the effect of the weakening of the $\mathbf{E}\times\mathbf{B}$ drift. However, this does not explain the substantial asymmetry of the changes across the magnetic equator in Figure 1c. Thus, we now look at the role of the neutral wind. The field line transport at 300 km (near the $F_2$ peak) in

this longitudinal sector (Figure 5a,b) is northward in both simulations, resulting in ions moved northward across the magnetic equator towards the northern EIA. The difference field *noobs* minus *hybma* in Figure 5c show a large positive (northward) difference in *noobs* just west of the longitude of the SAMI3 slice in Figure 4. This pattern of neutral winds illustrates that field-aligned transport in the *noobs* simulation is stronger than in *hybma*: the neutral winds around 300 km in *noobs* move ions across the magnetic equator and accumulate more ions just north of the magnetic equator setting up the north-south asymmetry

in the EIA difference field that is shown in Figure 1c around 90°E.





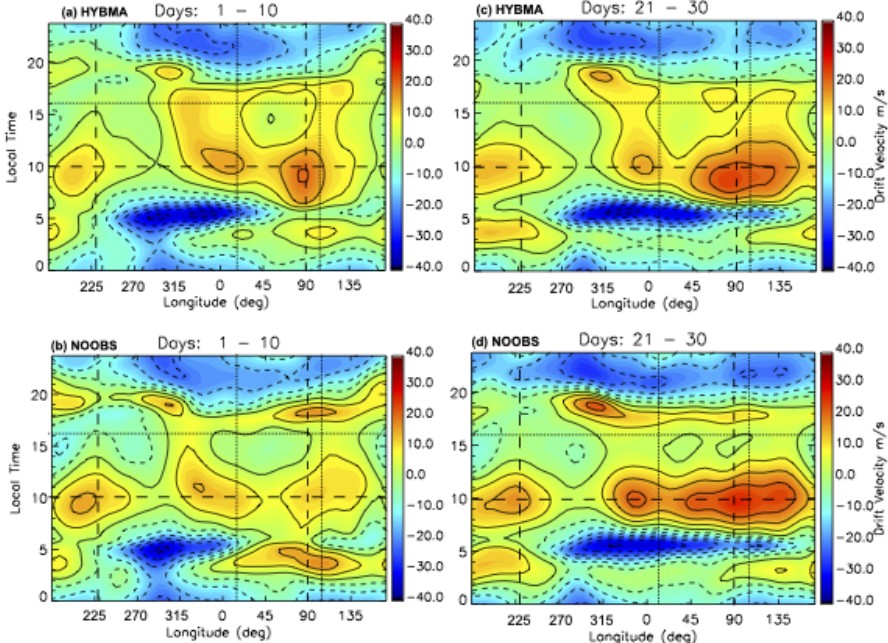

**Figure 3.** 10-day average of $\mathbf{E} \times \mathbf{B}$ ion drifts in m s$^{-1}$ obtained from the *hybma (a,c)* and *noobs (b,d)* simulations. *(a,b)* are the results of averaging during January 1–10, 2013; *(c,d)* show the drifts averaged during January 21–30, 2013. The dash black lines identify the approximate longitude (vertical lines) of the morning cases examined for 10:00 LT (horizontal line) and discussed in Figure 1. The dot black lines identify the approximate longitude (vertical lines) of the afternoon cases examined for 16:00 LT (horizontal line) and discussed in Figure 6 and Figure 8.

### 3.2   Case 2: 16:00 LT - dynamically active

We now turn our attention to the afternoon hours, with Figure 6 showing the $N_mF_2$ at 16:00 LT. This case illustrates another instance where the *noobs* and *hybma* simulations differ. At this LT, the $N_mF_2$ peaks broadly between 315°E to 90°E longitude in the northern hemisphere. The $N_mF_2$ in excess of $250 \times 10^4 cm^{-3}$ is broader and more prominent in *hybma* (Figure 6a), and

the minimum of $N_mF_2$ at the magnetic equator extends over more longitudes and is deeper in *hybma* than in *noobs* (Figure 6b).

Referring again to Figure 3(a,b) we notice that in the mid-afternoon (16:00 LT) around 15°E longitude the meridional $\mathbf{E} \times \mathbf{B}$ drift is stronger in *hybma* when compared to *noobs*. Therefore, in the simulation with MA observations, the electric field is more eastward (i.e., positive) when MA observations are used, resulting in stronger uplift of ions toward the F-peak. Figure 7 shows

the magnetic field-aligned component of the neutral wind at 300 km as a function of location at 16:00 LT. Note that the *noobs*



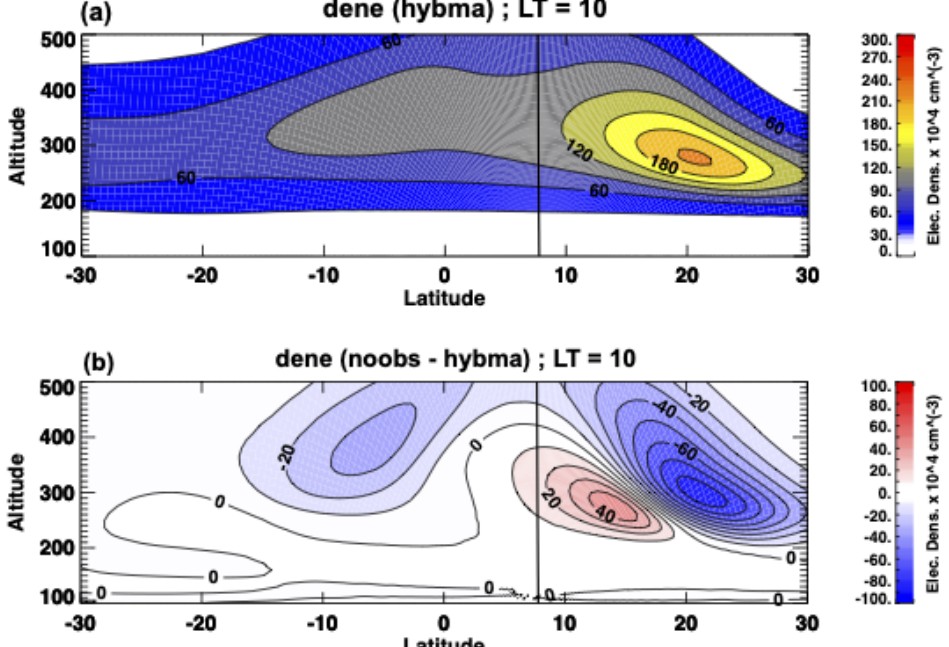

**Figure 4.** Electron density (dene) at 10:00 LT in units of $10^4 \text{cm}^{-3}$ along a SAMI3 slice with longitude at the equator $\sim 90°$, averaged for the same period as in Figure 1. The vertical black line identifies the magnetic equator. *(a)* electron density in *hybma*; *(b)* difference *noobs* minus *hybma*.

run has a stronger northward component along the SAMI3 slice crossing over Africa ($\sim 16°$) north of the magnetic equator, and a weaker northward component to the south of the magnetic equator. Thus, the *noobs* run has a larger neutral wind magnitude pushing the plasma northward along the field lines and a weaker fountain effect counteracting the interhemispheric transport. The difference in field parallel transport around the African continent SAMI3 slice, coupled with the different strength of the fountain circulation, explain the hemispheric asymmetry seen in the $N_mF_2$ at these longitudes between the two simulations.

### 3.3   Case 3: 16:00 LT - dynamically quiet

The case studies discussed in Secs. 3.1 and 3.2 illustrate the ionospheric response when the atmosphere is dynamically disturbed, such as during a SSW. As discussed in Section 1, anomalous ionospheric behavior is to be expected during these times. We demonstrate now that while this behavior is prominent during dynamically active times, the effect of MA observations is also detectable and statistically significant during dynamically quiet times.



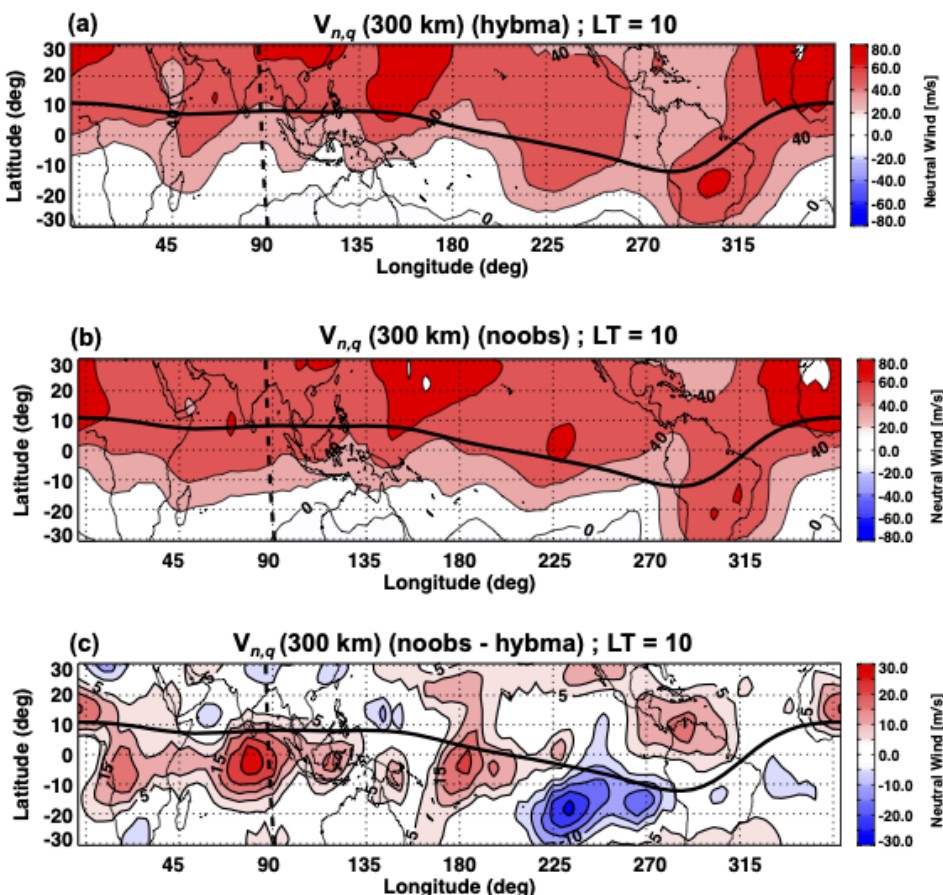

**Figure 5.** Neutral wind $V_{n,q}$ parallel to geomagnetic field lines at 300 km and 10:00 LT. Units are m s$^{-1}$; $V_{n,q}$ is averaged during 1–10 January 2013. The solid black line indicates the location of the magnetic equator, and the dashed line is the location of the SAMI3 slice crossing over the Indian Ocean. (a) $V_{n,q}$ for *hybma*  (b) $V_{n,q}$ for *noobs*; (c) $V_{n,q}$ difference *noobs* minus *hybma*.



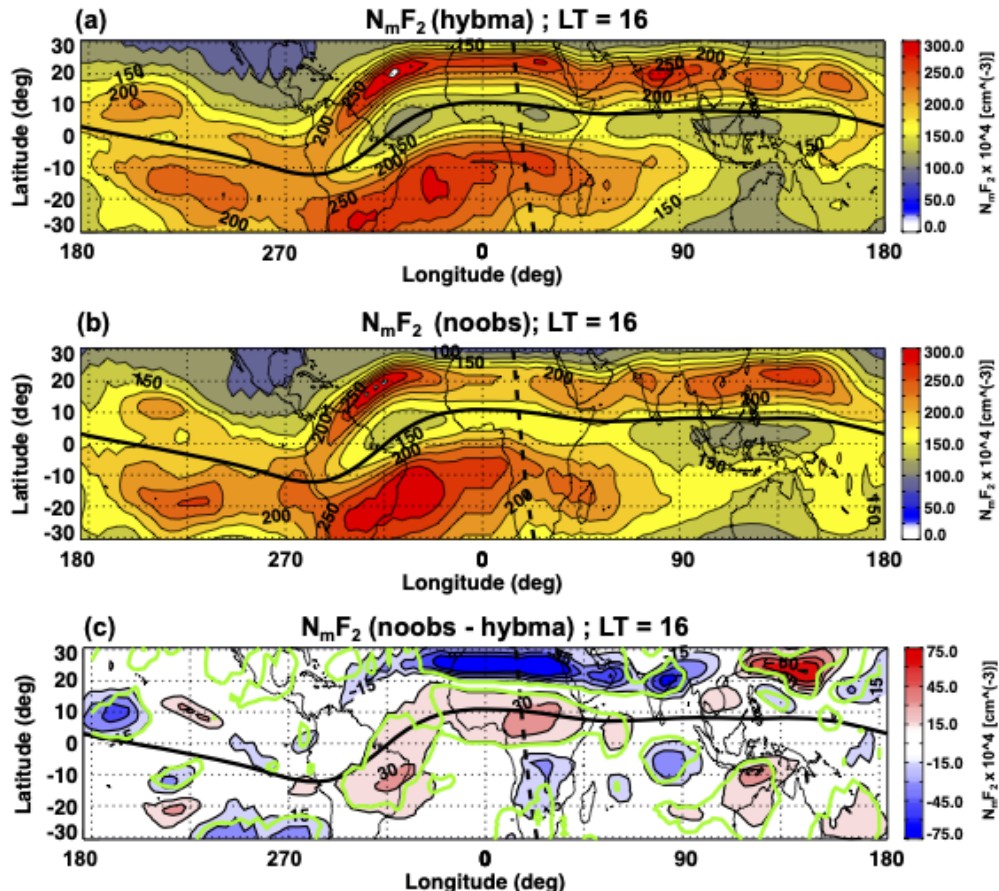

**Figure 6.** $N_mF_2$ in units of $10^4 \text{cm}^{-3}$ for all locations at 16:00 LT averaged for 1-10 January 2013: *(a) hybma*; *(b) noobs*; *(c)* difference *noobs* minus *hybma*. The oblique dash black line highlights the position of SAMI3 slice which crosses the geographic Equator at longitude $\sim15°$). The magnetic equator is described by a solid black line across longitude. Green contours in panel *(c)* identify the locations where the differences are statistically significant at least at 95% level using a t-test. Notice that compared to Figure 1, longitudes have been rotated to have $0°$E at the center.



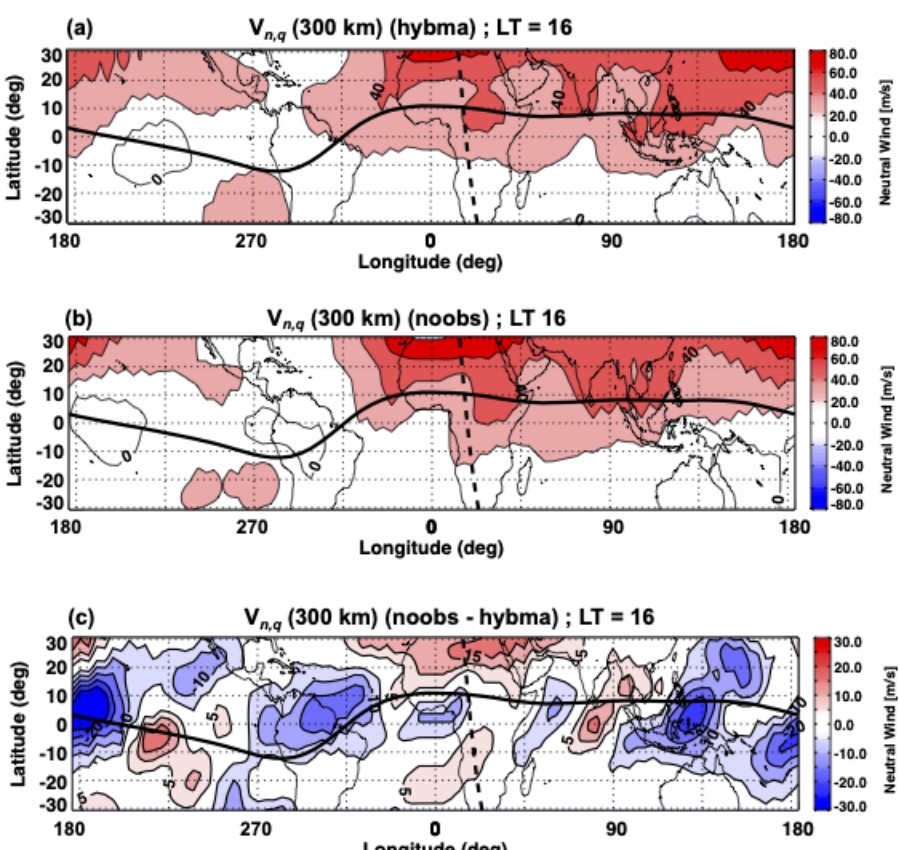

**Figure 7.** Neutral wind $V_{n,q}$ parallel to geomagnetic field lines at 300 km and 16:00 LT. Units are m s$^{-1}$; $V_{n,q}$ is averaged during 1–10 January 2013. The solid black line indicates the location of the magnetic equator, and the dashed line is the location of the SAMI3 slice crossing the geographic equator around longitude $\sim$16$^\circ$. (a) $V_{n,q}$ for *hybma* (b) $V_{n,q}$ for *noobs*; (c) $V_{n,q}$ difference *noobs* minus *hybma*.





Figure 8 is the equivalent of Figure 6 but averaged during the last 10 days of January (21–30 January 2013). This time period is well after the disturbance of the SSW on January 6. We chose the mid-afternoon (16:00 LT) because at this time the electron concentration is higher. During the dynamically quiet times of the end of the month, the largest differences between *hybma* and *noobs* shift over the Indonesian maritime region (dashed line). As in the case of the dynamically disturbed days shown in

Figure 6, there is a pronounced local minimum of $N_mF_2$ south of the magnetic equator in the *hybma* simulation, which results in a difference field with a positive $N_mF_2$ anomaly near the magnetic equator and negative $N_mF_2$ anomalies at sub-tropical latitudes. We note that these anomalies are statistically significant. Similar to the dynamically disturbed times earlier in the month, the anomalous $N_mF_2$ later in the month is somewhat asymmetric about the magnetic equator, indicating a continued role of the neutral wind in transporting ions across the magnetic equator (though to a lesser degree). Overall, Figure 8 shows

that the differences in $N_mF_2$ between the *hybma* and *noobs* simulations are not due to the characteristics of neutral dynamics during a SSW, but rather due to the lack of MA observations in the *noobs* simulation, which plays a role during all types of background conditions.

## 4 Discussion

This study uses two model experiments that couple a thermosphere climate model (WACCM-X), nudged with atmospheric

specifications (NAVGEM-HA), to an ionospheric model (SAMI3). The two experiments differ only in the observations that were used to generate the atmospheric specifications: the *hybma* experiment includes all observations up to the UMLT and the *noobs* experiment does not use MA observations above about 40 km. The goal of this study is to demonstrate that a potential future loss of MA observations will have significant negative consequences for our ability to predict the day-to-day variability of the neutral atmosphere and ionosphere, and is not only impactful for research on the neutral dynamics of the

lower thermosphere (as demonstrated by Sassi et al. (2020)).The results focused on the $N_mF_2$ at a fixed LT, as the peak electron density at the F-peak is a well-understood and appropriate diagnostic for ionospheric behavior. Using the $N_mF_2$ at fixed LT also allows an easier interpretation of the differences in terms of differences of non-migrating solar tides, as discussed by Forbes et al. (2006).

The results show that the difference of $N_mF_2$ between the two simulations is statistically significant (better than 95%

confidence level) at the selected locations and LT. In the morning (10:00 LT, Section 3.1), the largest differences between the two model runs is seen over the Pacific and Indian Oceans: a reduction of electric density in *noobs* near the magnetic equator is accompanied by an increase at adjacent latitudes in the Pacific Ocean, while the opposite signature is seen in the Indian Ocean region. The difference in behavior of $N_mF_2$ is found to be caused by a variation in the fountain effect when the MA measurements are not incorporated: stronger (weaker) uplift of ions over the Pacific (Indian) Ocean when MA observations are

removed. At different solar local times and geographic locations, the neutral winds introduce in both simulations a hemispheric asymmetry to the equatorial plasma distribution.

During the afternoon (16:00 LT, Section 3.2), the changes of $N_mF_2$ over the African continent are similar to those seen in the morning over the Indian Ocean. Examination of the changes along a magnetic meridian shows in fact that the afternoon

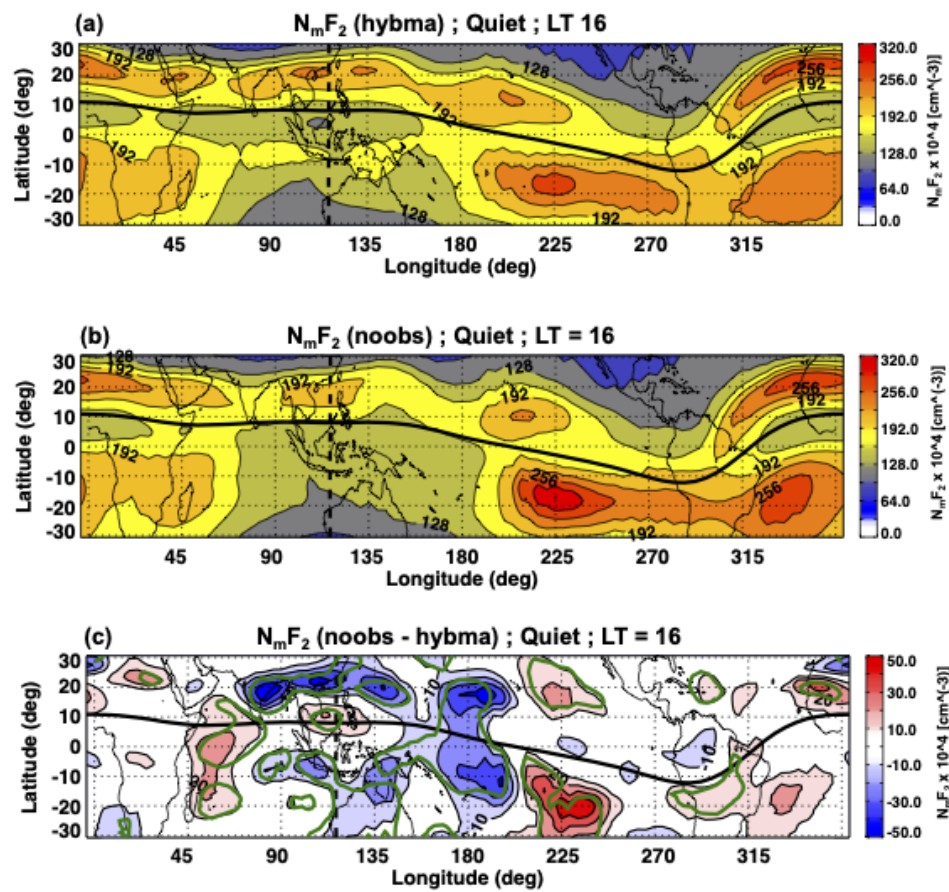

**Figure 8.** As in Figure 6, $N_mF_2$ in units of $10^4 \text{cm}^{-3}$ for all locations at 16:00 LT averaged for 21-30 January 2013: *(a) hybma*; *(b) noobs*; *(c)* difference *noobs* minus *hybma*. The dash black line highlights the position of SAMI3 slice crossing the geographic equator over the Indian Ocean at longitude ~117°. The magnetic equator is described by a solid black line across longitude. Green contours in panel *(c)* identify the locations where the differences are statistically significant at least at 95% level using a t-test.



anomalies are due to the decrease in the magnitude of the meridional $\mathbf{E}\times\mathbf{B}$ drift in *noobs* when the MA observations are
not considered. The magnetic field-aligned component of the neutral wind at 300 km (an altitude chosen as it is just below
the peak $h_mF_2$) shows that the longitudinal structure of the lower thermospheric neutral wind has also been affected by the
differences in MA forcing. This, in turn, changes the magnitude of the interhemispheric transport at this local time, affecting
the longitudinal structure of the $N_mF_2$.

The first two cases illustrated (Sections 3.1 and 3.2) include the occurrence of a SSW in the stratosphere, which is known to
produce anomalous behavior in the thermosphere and ionosphere; we therefore examined also a dynamically quiet case towards
the end of January 2013 (Section 3.3). Overall, the effect of removing MA observations from the atmospheric specifications
during a dynamically quiet period is consistent with the more dynamically disturbed cases, although showing a more muted
response. This indicates that a dynamically disturbed atmosphere is not required for MA observations to impact predictions of
ionospheric structure, although a disturbed state can mediate the magnitude of the impact.

The zonally articulated response in $N_mF_2$ in all cases due to the exclusion of MA observations indicates that there are
differences in the amplitude of non-migrating solar tides. The zonal structure of the difference field indicates prominent wave-3
or wave-4 modes at fixed LT, which is consistent with past findings. For instance, Sassi et al. (2020) showed that the simulation
without MA observations (i.e., *noobs*) produces much larger non-migrating tides DE3 and DE2. These solar tides are expected
to be visible in $N_mF_2$ as wave-4 and wave-3 patterns when plotted at fixed LT.

Finally, we compare the results of these runs to TEC observations made at a nearby longitude slice just eastward of the one
shown in Figure 6 at 110° apex longitude. This longitude slice lies over portions of Africa and Europe that are sufficiently
instrumented to provide coverage of the EIA. For each day of the model simulations, the Vertical TEC (VTEC) was selected
within 3° (for the model runs) or within 5° (for the observational data) and between 15:45 and 16:15 LT. Then, the mean VTEC
was calculated at a 1° latitude resolution. Figure 9*(right)* shows the geographic longitude of the observations and nearby model
data used in our analysis. Figure 9*(left)* shows the result of this averaging process, with the *noobs* run marked by navy circles,
the *hybma* run marked by light blue squares, and the observed VTEC from MIT Haystack (Rideout and Coster, 2006) marked
by slate blue triangles.

Figure 9 shows a day where the two model runs have differing EIA structures, and neither of them fully agree with the
observed data. The first level of disagreement is the bias between the model and observations. Depending on the neutral
atmospheric composition and density, the SAMI3 model tends to overestimate the ionospheric plasma density. When MSIS
is used to provide the background density, as is the case in Figure 9, this difference is on the order of ∼10 TECU near the
equator. These differences are most obvious in the VTEC, as it is vertically integrated and exacerbates the problem. Figure 9
also shows the three potential configurations of the EIA encountered during this study: no dual EIA peak (No, shown by the
*noobs* data), a higher northern EIA peak (North, shown by the observational data), and a higher southern EIA peak (South,
shown by the *hybma* data). We will compare the state of the EIA between the observations and the model runs by determining
the times when the modeled and observed EIA had the same configuration.

Tables 1 and 2 show the number of days in the month of January 2013 where each model run had a particular EIA con-
figuration, and how this compares to the observed EIA configuration. Comparing the two tables shows that introducing MA



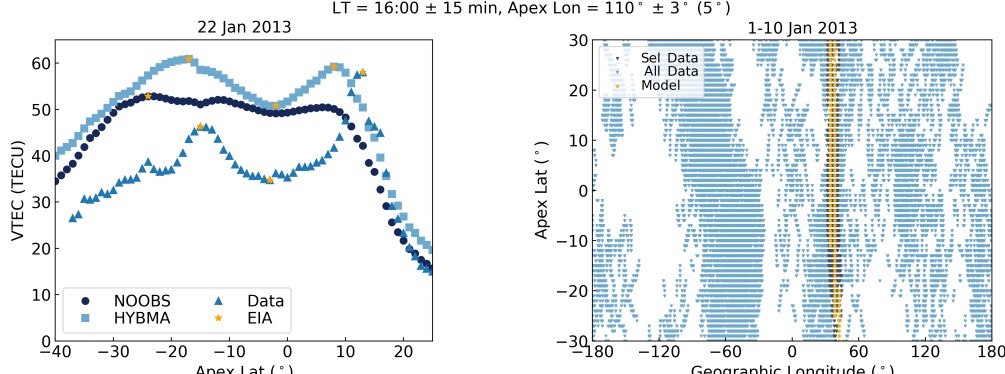

**Figure 9.** *(left)* Mean VTEC as a function of magnetic apex latitude at 16:00 LT at 110° magnetic apex longitude on 22 January 2013. The *noobs* results are marked by navy circles, the *hybma* results are marked by light blue squares, and the observation data are marked by slate blue triangles. The EIA peak and trough locations (or only a central peak) are marked by orange stars. *(right)* Geographic longitude and magnetic latitudes of all available data (light blue circles), and the selected data (dark blue circles) used to assemble the statistics of Tables 1 and 2; the corresponding model data (orange circle) are also shown. The panel was constructed for 1–10 January 2013, and provided here for reference to show that observational data is broadly located at ∼36°E.

observations in the *hybma* run improves the agreement between the observations and the model. This is occurs specifically with
an increase of the number of days both the model and observations show an EIA with a dominant northern peak: *noobs* had six days in this EIA configuration, three of which agreed with the observations and *hybma* had 13 days in this EIA configuration, eight of which agreed with the observations. As this is the dominant state in the observations (14 days in total), it greatly improves the overall agreement. This area of change confirms that the improved lower atmospheric boundary does a better job specifying the upper thermospheric wind, which drives the EIA hemispheric asymmetries.

**Table 1.** EIA configuration agreement between *noobs* and observed VTEC at 110° magnetic apex longitude, 16:00 LT.

| *noobs* | Observed | | | |
|---|---|---|---|---|
| | South | North | No | Marginal Total |
| South | 3 | 10 | 6 | 19 |
| North | 2 | 3 | 1 | 6 |
| No | 0 | 1 | 5 | 6 |
| Marginal Total | 5 | 14 | 12 | 31 |

# 5 Conclusions

The results of this study support the thesis presented in the introduction: the absence of MA observations has important and statistically significant consequences on our ability to predict the structure and variability of the ionosphere and upper



**Table 2.** EIA configuration agreement between *hybma* and observed VTEC at 110° magnetic apex longitude, 16:00 LT.

| *hybma* | Observed | | | |
|---|---|---|---|---|
| | South | North | No | Marginal Total |
| South | 2 | 4 | 5 | 11 |
| North | 3 | 8 | 2 | 13 |
| No | 0 | 2 | 5 | 7 |
| Marginal Total | 5 | 14 | 12 | 31 |

atmosphere. The impact of the MA observations on the E-region conductivity and thermospheric neutral winds demonstrate that this region is an essential piece that must be understood to fully grapple with interactions between the neutral atmosphere

and the ionosphere. At lower latitudes, multiple types of interactions between the thermosphere and ionosphere can lead to ion transport that affects the F-region peak density distribution. Model runs driven with incomplete MA observations may have anomalous neutral wind at E-region altitudes that lead to an aberrant fountain effect through electrodynamic interactions. They may also have atypical F-region neutral winds that cause abnormal hemispheric electron density distributions, such as a higher electron density in one hemisphere or a decrease in the seasonal EIA asymmetries.

Strictly speaking, these results apply only to the modeling system we have examined. Other modeling systems (for example, systems that assimilate ionospheric data) may see different impacts on the ionospheric state. It is possible that assimilating sufficient data from appropriate ionospheric and thermospheric data sets could mitigate the need for MA observations for ionospheric predictions. To the best of our knowledge, this is the first study that attempts to investigate how a future lack of observations in the MA potentially impacts the upper atmosphere and ionosphere. Given the importance for civilian and military

applications of high fidelity predictions in the upper atmosphere and ionosphere, it is necessary to continue and expand these studies, including examining results from other modeling systems.

*Code and data availability.* SD-WACCM-X simulation output and NAVGEM-HA atmospheric specifications are archived at the Naval Research Laboratory. The source code and forcing datasets of the SD-WACCM-X is available from the NCAR website http://www.cesm.ucar.edu/. GPS TEC used in this study is publicly available from the Madrigal database and was converted into magnetic coordinates using apexpy

(van der Meeren et al., 2023).

*Author contributions.* FS conceptualized this study and its methodology, led the writing, and performed the model analysis. AGB led the physical analysis and contributed to both the writing and model analysis. SEM contributed to the physical and model analysis, as well as editing. JLT performed model runs and contributed to the writing. JPM contributed to the writing and model analysis.



*Competing interests.* The authors declare that no competing interests are present.

*Acknowledgements.*  FS, AGB, and SEM acknowledge the support of the Office of Naval Research. This work is supported in part by a grant of computer time from the DOD High Performance Computing Modernization Program at the US Navy DOD Supercomputing Resource Center (NAVO).

GPS TEC data products and access through the Madrigal distributed data system are provided to the community by the Massachusetts Institute of Technology under support from U.S. National Science Foundation grant AGS-1242204. Data for the TEC processing is provided

by the following organizations: UNAVCO, Scripps Orbit and Permanent Array Center, Institut Geographique National, France, International GNSS Service, The Crustal Dynamics Data Information System (CDDIS), National Geodetic Survey, Instituto Brasileiro de Geografiae Estatística, RAMSAC CORS of Instituto Geográfico Nacional del la República Argentina, Arecibo Observatory, Low-Latitude Ionospheric Sensor Network (LISN), Topcon Positioning Systems, Inc., Canadian High Arctic Ionospheric Network, Institute of Geology and Geophysics, Chinese Academy of Sciences, China Meterorology Administration, Centro di Niveau des Eaux Littorales Ricerche Sismogiche, Système

d'Observation du (SONEL), RENAG : REseau NAtional GPS permanent, and GeoNet—the official source of geological hazard information for New Zealand.





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
