# Peer review of "On the importance of middle atmosphere observations on ionospheric dynamics using WACCM-X and SAMI3"

_EGUsphere, 2023_

## Author Response (AR1)

We thank the reviewer for their comments. We have made the suggested corrections to the text.

Line 131 (now line 136): Fixed reference authorship

Line 311 (now line 328): Made recommended change to text

We thank the reviewer for their comments.  We have made many of the suggested corrections to the text, all points are discussed below.  When referring to line numbers below "line X" refers to lines in the original document while "new line Y" refers to line numbers in the updated document.

General points:

"The author focus on specific LT and averages, but it would bemore meaningful to quantify the change in day-to-day variability between the two cases (The authors mention several time the importance of day-to-day variability). While the text mentions day-to-day variability the study itself focuses on variability in general during the specific time periods. The study would gain if ability to capture the day-to-day variability would be added."

- We agree that this would be an interesting study, but is beyond the scope of the current paper.

"If the authors want to demonstrate the value of adding middle atmospheric observations to better specify the ionospheric variability, it is surprising that the focus is on 10 LT. The authors should either motivate it or add later local times especially in the afternoon."

- We focused on two different local times, 10:00 LT and 14:00 LT. These two times were chosen because of the observed changes in the ionosphere.  Motivation for these times can be found in the new manuscript on lines XXX and XXX.

"The condition during the time period needs to be described in one short section at the beginning (inc. F10.7, PW evolution & SSW peak, geomagnetic activity). Would the results be different without the change in PW activity. How general is the result?

Wasn't there some minor geomagnetic activity as well during the time period?"

- This was included in the start of Section 3, and is now in the first paragraph of Section 3.  The atmospheric behavior for the time periods used in this study is also discussed in detail in the paper referenced at the end of this section.

"Please make the longitudinal range the same in the plots, so that 0 deg is in the same location e.g. Fig 1 and Fig 3."

- The longitude ranges differ in the figures so that the longitudes of interest can be easily observed.  While we appreciate that it is annoying to have figures with different longitude ranges, we felt it was more important to chose ranges that ensured the selected longitudes of interest were not split across the edge of the figure.

Specific comments:

Line 12/new line 12-13: This line was changed to "variability induced by wind-dynamo coupling through electric conductivity and ion-neutral interactions in the upper thermosphere." to clarify the intention of the text.

Line 20/new line 20: Changed to "the short-term variability in the E-region"

Lines 27-28/new lines 27-28: Added "with respect to the long-term mean (calculated using data from 1967-1989)"

Line 35/new line 36: Added "When studied at a fixed local time (LT)," to provide the specific context for this sentence.

Line 52/new line 52: changed "forecasting skills" to "predictions"

as suggested.

Line 54/new line 54: clarified by adding: "global root mean square error in zonal wind"

Line 62/new line 62: made suggested change.

Line 64/new line 64: added "during boreal winter"

Line 66/new line 66: CITS was first defined in the previous paragraph (new line 48).

Line 90-92/new line 92-93: clarified by adding a sentence: "Due to the sparseness of observations, especially in the critical UMLT, analysis fields at a cadence higher than 6 hours are indistinguishable from intermediate forecasts, and we augmented the 6-hour analysis with 3-hourly forecasts; this approach has also been used in (Sassi et al. 2020)."

Line 99/new line 101: This would be a different study that would require the use of WACCMX+DART.  While this alternative approach would also likely be interesting, our method is more adaptable to a mechanistic study like the one presented here. Using WACCMX+DART would not be mechanistic (allowing us to easily see the theoretical effect on the ionosphere driven by MA data assimilation), but an interactive study that makes it more difficult to isolate the driving mechanisms.

Line 105/new line 107: This is not possible to do because of the model infrastructure.  NAVGEM uses a spectral triangular formulation (hence, T119) and WACCM is is a grid-point model. Attempting to use the same formulation of units and grid resolution for both models would be incorrect in their implementations.

Line 114/new lines 116-118: I am not sure what kind of "zonal convection" the reviewer is referring to. The SAMI3 description was updated to add: "…a two-dimensional model of the ionosphere that handles plasma dynamics and chemical evolution along magnetic field lines (varying in latitude and altitude). SAMI3 extends SAMI2 by adding the longitudinal dimension, which includes zonal transport." If the reviewer is referring to high-latitude plasma convection, then this is handled by driving SAMI3 with Weimer.

Line 122 /new lines 125-128: provided a more thorough explanation of modified apex coordinates.

Line 125: Further details about the standard SAMI3 set up are available in the references.

Line 129/new line 133: The software allows for different way to couple the ionosphere to the thermosphere. For this particular study, we are using a one-way coupled configuration for practical reasons, but the software developed at NRL is also capable of supporting two-way coupling. We changed the wording to say: "The software infrastructure that extends SAMI3 to allow either one-way or two-way coupling with the atmosphere". We also updated the wording in the subsequent paragraph to make it clear that we are using a one-way coupled set-up, and why we chose to do so.

Line 139/new line 143-149: reworded introduction entirely, taking this into account.

Caption Fig 4: Adjusted wording to say: "along a SAMI3 slice with a longitude of ~90˚E at the geographic equator"

Line 156/new line 171: changed wording to "more complex longitudinal structure"

Line 164: It may be possible that there are changes in the phases in the migrating tides that are affecting the ionosphere. However, our approach is to examine the impacts in a local time framework. This makes the phase changes of the migrating tides unimportant, since they can't be observed in this frame of reference.

Figure 1: 10LT and 16LT are used as representative of a complex behavior and exemplify the point of our study: lack of MA observations impacts the ions distribution and transport. The reviewer is correct in pointing out that more advanced studies can be done, but this is a a simple exemplification of many other facets. It is also important to examine the impact of MA observations at different local times. While this selection of case studies is limited, we think it is important to focus on more than just one LT region.

Line 169/new lines 186-187: changed to: "Once the vertical ExB drifts lift the plasma to magnetic field lines with higher apex altitudes, the change in the plasma pressure gradient along these longer field lines will cause the ions to diffuse downwards along the magnetic field lines to lower altitudes and higher latitudes"

Line 171: We also look at the ExB drift directly. See Figure 3 and Line 200 (new line 200-210).

Line 180: The reviewer may be correct, but for this study it doesn't matter: the goal of the study is not to investigate the detailed processes operating at this time; it is a sensitivity study to demonstrate that the lack of MA observations is impactful on the ionospheric properties. To ensure this is clearer, we added "at this time and location" to the end of this sentence.

Sect 3.2: This wording refers to the presence or absence of a SSW. This is made more clear now by rewriting the introduction to the case studies in the first paragraph of Section 3.

Line 254 and elsewhere: Replaced "day-to-day" with "intra-day" throughout the text.

Line 255/new line 271: It would be incorrect to remove one parenthesis, since two are needed for closure.

Line 274/new line 290: The SSW is first mentioned on Line 151. The new rewording now brings it up in the first paragraph of Section 3.

Line 276/new line 292: Both an SSW period and a period without SSW were examined to see if the influence of MA observations would be important during extreme times, normal times, or not be important at all. This is now clarified in the first paragraph of Section 3.

Line 281/new line 296: There may be other things, but the MA observations specifically affected the non-migrating solar tides, as is possible due to the nature of this controlled mechanistic experiment. This is why we are talking about their impacts in this paragraph.

Line 284/new line 300: Added reasoning behind discussing these wave patterns in particular to the text.

Figure 9: We agree with the reviewer, and the text states that the NOOBS example shows an instance without the EIA present. The figure caption also says that in the absence of an EIA, the start will mark the peak density in this region. The figure caption

has been further clarified to avoid confusion.

Table 1 and 2: Adjusted the wording in new lines 310-318 more clearly explain that we are examining the low-latitude VTEC distribution by grouping data into three types of EIA configurations: no EIA, an EIA with a higher northern peak, and an EIA with a higher southern peak.

Line 313/new line 330: Added text relating the E-region conductivity to the transport caused by the ExB drift.

Line 320: :)

Discussion: We decided not to do this, because we are specifically interested in the way the MA drives the F-region ionosphere.  Because the MA is below the ionosphere, we do not expect two way coupling to change the conclusions we found here.

[revised manuscript text omitted]